# Clinical Characteristics and Outcome of MDR/XDR Bacterial Infections in a Neuromuscular Semi-Intensive/Sub-Intensive Care Unit

**DOI:** 10.3390/antibiotics11101411

**Published:** 2022-10-13

**Authors:** Arta Karruli, Alessia Massa, Lorenzo Bertolino, Roberto Andini, Pasquale Sansone, Salvatore Dongiovanni, Maria Caterina Pace, Vincenzo Pota, Emanuele Durante-Mangoni

**Affiliations:** 1Department of Precision Medicine, University of Campania ‘Luigi Vanvitelli’, Via de Crecchio 7, 80138 Napoli, Italy; 2Unit of Infectious and Transplant Medicine, AORN Ospedali dei Colli-Monaldi Hospital, Piazzale Ettore Ruggieri, 80131 Napoli, Italy; 3Department of Woman, Child and General & Specialized Surgery, Section of Anesthesiology, University of Campania ‘L. Vanvitelli’, Piazza Miraglia, 80138 Napoli, Italy; 4Centro Clinico NEMO, AORN Ospedali dei Colli-Monaldi Hospital, Piazzale Ettore Ruggieri, 80131 Napoli, Italy

**Keywords:** neuromuscular disease, MDRO/XDRO, colonisation, invasive device, carbapenem

## Abstract

(1) Background: The aim of this study was to assess the clinical and microbiological characteristics of multidrug-resistant infections in a neuromuscular semi-intensive/sub-intensive care unit; (2) Methods: Retrospective analysis on data from 18 patients with NMD with proven MDRO/XDRO colonisation/infection from August 2021 to March 2022 was carried out; (3) Results: Ten patients were males (55.6%), with a median age of 54 years, and there were fourteen patients (77.8%) with amyotrophic lateral sclerosis. All patients had at least one invasive device. Ten (55.6%) patients developed MDRO/XDRO infection (with a median time of 24 days) while six (33.3%) were colonised. The Charlson comorbidity index was >2 in both groups but higher in the infected compared with the colonised (4.5 vs. 3). Infected patients were mostly females (seven patients) with a median age of 62 years. The most common pathogens were *Acinetobacter baumannii* and *Pseudomonas aeruginosa,* infecting four (28.6%) patients each. Of eighteen infectious episodes, nine were pneumonia (hospital-acquired in seven cases). Colistin was the most commonly active antibiotic while carbapenems were largely inactive. Eradication of infection occurred in seven infectious episodes (38.9%). None of those with infection died; (4) Conclusions: MDRO/XDRO infections are common in patients with neuromuscular diseases, with carbapenem-resistant non-fermenting Gram-negative bacilli prevailing. These infections were numerically associated with the female sex, greater age, and comorbidities. Both eradication and infection-related mortality appeared low. We highlight the importance of infection prevention in this vulnerable population.

## 1. Introduction

Increased spread of antimicrobial resistance over the past years has been translating into growing morbidity, mortality, and healthcare costs of affected patients [1,2].

Infections due to multidrug-resistant organisms (MDRO) account for 31.6% of hospital infections [3] and are a major threat in intensive care unit (ICU) settings, where their prevalence may reach up to 50% [4,5,6]. Indeed, factors associated with an increased MDRO infection risk include characteristics largely seen in ICUs such as critical illness, large-spectrum antibiotic administration, invasive procedure performance (mechanical ventilation, endotracheal tubes, central venous access, urinary catheters, feeding tubes, etc.), and also hospital/healthcare exposures related to prolonged hospital stay [7,8,9,10,11,12]. Therefore, patients with chronic illnesses who require frequent hospitalisations and/or antibiotic treatments as well as invasive procedures are at risk of developing MDRO infection along with various complications thereof. A special group of patients at such a high risk could be represented by patients with neuromuscular disorders (NMDs).

NMDs include a broad spectrum of inherited or acquired diseases characterized by lesions that may involve motor neurons, the neuromuscular junction, and/or the skeletal muscle [13,14].

Respiratory muscle weakness is a common feature among these patients, leading to impaired airway clearance [15,16,17] and hypoventilation. Development of respiratory failure is common and may associate with bulbar muscle (mouth and throat muscles responsible for speech and swallowing) hypotonia, concomitant anatomical malformations (scoliosis/rigid spine), and a decrease in central respiratory drive [18]. All these factors may place the patient in the need of non-invasive ventilation (NIV) in the intermediate stages of the condition, followed by invasive ventilation (e.g., tracheostomy) in keeping with the progression of the disease [17,19].

In addition to a weak cough with an accumulation of phlegm and mucus in the throat and/or nasal passage, recurrent aspiration leading to frequent respiratory tract infections, bulbar muscle weakness can also lead to swallowing problems requiring frequent hospitalisations for feeding tubes and intravenous catheter (central/peripheral) placement [17]. Specialized semi-intensive care units are increasingly dedicated to the care of NMD patients.

These patients have a great host of various risk factors for MDRO infections, such as frequent hospitalisations, need for invasive procedures (feeding tubes, tracheostomy, central venous catheters, urinary catheters), and frequent antibiotic treatments. However, there is little if any literature data regarding MDRO infection in NMD patients. Therefore, we carried out this study with the aim to assess the clinical and microbiological characteristics of MDRO infections in a neuromuscular semi-intensive/sub-intensive care unit.

## 2. Materials and Methods

### 2.1. Study Design

This was a retrospective, observational study. All patients who underwent hospitalisation in the neuromuscular semi-intensive/sub-intensive care unit of the V. Monaldi Hospital in Naples, Italy, between August 2021 and March 2022 were included in this study. Data regarding the clinical characteristics of patients, neuromuscular disease, comorbidities, as well as clinical features, microbiological diagnosis, and outcomes of infectious episodes, were recorded. Infections with a microbiological diagnosis with clinical signs of infection coupled with biochemical inflammatory parameter elevation, occurring during hospitalisation, were considered for this analysis. Surveillance microbiological sampling of blood, urines, airways, and surgical wounds were performed in all patients at least once weekly, and additional cultures were performed as dictated by patient conditions. Data collection was approved by the Ethics Committee of the University of Campania ‘Luigi Vanvitelli’ and the AORN Ospedali dei Colli (protocol no. AOC/28408/2021).

### 2.2. Definitions

Antimicrobial susceptibilities of microbial isolates were performed using the Vitek-2 system and the AST-GN card (bioMérieux, Marcy l’Etoile, France). Values were interpreted according to breakpoint table for interpretation of MIC values and zone diameters (European Committee on Antimicrobial Susceptibility Testing, 2021) [20].

Infections were classified as being due to “multidrug-resistant” (MDR), “extensively drug-resistant” (XDR), or “pan-drug-resistant” (PDR) pathogens, in accordance with the definitions of Magiorakos et al. Accordingly, “MDR bacteria were defined as bacteria that are non-susceptible to at least one antimicrobial in three or more antimicrobial classes, XDR was defined as non-susceptibility to at least one antimicrobial agent in all but two or fewer antimicrobial classes” [21].

Infections were diagnosed based on the current US Centers for Disease Control and Prevention National Healthcare Safety Network criteria [22]. Patients who only showed MDR bacterial colonisation (rectal/nasal carriers) were not included among patients with infection.

The infectious syndromes number was calculated based on the number of infection types of each patient; therefore, if patients developed more than one type of infection in the same or different time frame, the total number of patients and infectious syndromes do not match exactly. Infections due to extended-spectrum beta-lactamase (ESBL)-producing Enterobacteriaceae which did not show resistance against other groups of antibiotics were not included among MDR/XDR infections due to the endemic spread of these microorganisms in our clinical setting. Eradication of infection was defined as negativity of follow-up cultures coupled with clinical and biochemical parameter improvement. For infection prevention, in our centre, we use chlorhexidine bathing along with substitution and sterile handling of catheters.

### 2.3. Analysed Variables

For each patient, we collected general clinical data, hematochemical parameters, and treatments received.

Among general clinical data we considered age, sex, comorbidities, and hospitalisation in the 90 days prior to the current one. Comorbidities were assessed by means of the Charlson comorbidity index. We also recorded the presence of venous catheters, feeding tubes, and tracheostomy.

Hematochemical parameters were collected on hospital admission, infection episode onset, and hospital discharge/decease; these included white blood cell count, platelet count, creatinine, bilirubin, and albumin.

Regarding antimicrobial treatment administration, we analysed the antibiotic therapy given for each infectious episode.

Outcomes considered were eradication of infection, mortality, and relation of mortality with MDRO infection.

### 2.4. Statistical Analysis

Descriptive statistical analysis was performed on data obtained at the time of hospital admission and MDRO isolation. Numerical variables were expressed as median and interquartile range (IQR), while categorical variables were expressed as number and percentage.

## 3. Results

Eighteen patients with a positive culture for MDR/XDR pathogens were included, whose baseline clinical features are presented in Table 1. The majority of patients were males (10 patients) with a median age of 54 years (IQR 45–69). Amyotrophic lateral sclerosis was the most common primary NMD in 14 patients, and hemiplegia was the most common complication/comorbidity in 15 patients. A previous hospitalisation occurred in 12 patients. All patients underwent at least one invasive procedure, with parenteral nutrition being the most common in 14 of them, followed by tracheostomy in 13.

Ten of eighteen patients (55.6%) developed an infection due to documented MDRO/XDRO (Figure 1), whereas six (33.3%) patients who were colonised did not develop an infection and two (11.1%) patients with colonisation developed an infection but with an unknown pathogen and, therefore, were excluded from the infected patients. The gut was the most common source of colonisation and Klebsiella pneumoniae was the most common pathogen among patients with colonisation, whether they developed an infection or not (Figure 1).

According to MDRO/XDRO infection development, we divided patients into two groups: (a) MDRO/XDRO-infected patients and (b) MDRO/XDRO-colonised but not infected patients (Table 1). No major differences were seen between the groups. However, the median age was numerically higher in the infected group, with a median of 62 (IQR 43–70) vs. 51 (IQR 43–61). Females predominated in the infection group, accounting for 70% of cases, whilst males predominated in the colonisation group (83.3%). Furthermore, the Charlson comorbidity index was higher in the infection group, with a median of 4.5 (IQR 2–6) vs. 3 (IQR 2–4.7) in the colonised. Although all patients underwent at least one invasive procedure, when analysed individually, most of the invasive procedures were more commonly performed in infected patients. Regarding hematochemical parameters, the increase in white blood cells (WBC) and decrease in albumin during hospitalisation was most common in the infected group (four of ten patients vs. zero, and five of ten patients vs. zero, respectively). None of the patients who developed an infection died during hospitalisation. In contrast, one patient in the colonisation group died, although the reason of death was not related to colonisation.

Characteristics of MDRO/XDRO infections are shown in Table 2. Of the ten patients who developed an infection, three were previously colonised. Of these, one patient developed an infection due to the same pathogen and two patients due to other pathogens. Overall, 10 patients developed 18 infection episodes due to 14 pathogens. Four patients had more than one pathogen isolated. The most common infectious syndrome was pneumonia in nine infectious episodes (in seven of nine hospital-acquired), followed by complicated urinary tract infection in four, skin and soft tissue infection in three, and bloodstream infection in two. Gram-negative pathogens predominated, accounting for 12 (85.7%) of the total isolates, with the most common pathogens being Acinetobacter baumannii and Pseudomonas aeruginosa (each four of the total isolates). The median interval from hospital admission to culture positivity was 24 days. Eradication of infection occurred in seven (38.9%) infectious episodes whereas pathogens persisted in eight (44.4%) and information regarding follow-up culture was missing in three (16.7%). Antibiotic treatments are described in Table 2. The median duration of treatment was 12 days with an IQR of 7 to 21 days.

Antibiotic susceptibility is described in Appendix A. Most of the isolates were shown to be resistant to most of the antibiotics, in particular meropenem. Interestingly, new antibiotics such as Ceftolozane/Tazobactam and Ceftazidime/Avibactam were found to be inactive in vitro against some of the Gram-negative isolates Most of the isolates were susceptible to colistin. However, the three infectious syndromes treated with colistin did not eradicate the infection.

Combination therapy was used in three patients, specifically: Ampicillin/Sulbactam + Gentamicin; Cefiderocol + Teicoplanin; and Piperacillin/Tazobactam + Ceftriaxone. Other antibiotics were used in monotherapy. Regarding the use of new antibiotics such as Cefiderocol, one case treated with Cefiderocol eradicated the infection whereas one did not.

## 4. Discussion

Data in the literature regarding MDRO/XDRO infection in patients suffering with NMD and requiring hospitalisation, in particular, in a semi-intensive/sub-intensive care setting are scarce, with only a few studies describing pneumonia as one of the most common complications of this group of diseases [23,24]. Indeed, mortality due to pneumonia in amyotrophic lateral sclerosis and adult-onset myotonic dystrophy is shown to be near 30% [25,26]. Pneumonia was also the most common infectious syndrome in our cohort of patients, with 50% of them developing pneumonia during hospitalisation. Not only was this group of patients predisposed to develop pneumonia, but hospital-acquired pneumonia was also the most common hospital infection in a point prevalence study carried out in 28 European countries for the period 2016–2017 [3]. The five most common pathogens causing hospital-acquired pneumonia are *S. aureus*, *Pseudomonas* spp., *Acinetobacter* spp., *Escherichia* spp., and *Klebsiella* spp., which cause nearly 80% of all episodes [27], with *Staphylococcus aureus* being the main causative pathogen [28]. In contrast, in our cohort of NMD patients, only two of nine episodes of pneumonia were attributed to *Staphylococcus aureus* (22.2%), whereas three episodes were attributed to *Pseudomonas aeruginosa* or *Acinetobacter baumannii* (33% each), and one (11.1%) to *Klebsiella pneumoniae*. Even though, in our cohort of patients, *S. aureus* was not the most common pathogen, our findings are in line with the prevalence of *Staphylococcus aureus* as a causative agent of hospital pneumonia in Europe, which is 23% [28].

In our experience, 77.7% of pneumonia episodes were attributed to Gram-negative pathogens and, since aspiration pneumonia is common in NMD patients, this is a plausible result [24].

In terms of demographic characteristics, our cohort was made up of slightly more males (55.6%), who have the highest rate of amyotrophic lateral sclerosis [29]. In our cohort, males also had the higher percentage of colonisation in line with other studies which suggest the male gender to be the most vulnerable to MDRO/XDRO acquisition [8,30,31]. However, females prevailed in the infection group of our cohort [32].

Our cohort of patients had a high burden of comorbidities with a median Charlson comorbidity index of 3.5 and a median of 4.5 in the group of MDRO/XDRO-infected patients. A study by Laudisio et al. showed that among patients with urinary tract infections, the ones who developed a MDRO/XDRO-related episode had a higher CCI [33].

The median time from hospital admission to culture positivity was 24 days, longer than evidenced in a previous study of MDRO infections in COVID-19 at our centre, which was eight days [34].

In our experience, MDRO/XDRO infections were also a common complication in other subgroups of patients suffering from chronic illness such as heart transplant patients with Gram-negative pathogens prevailing as published by our centre [35]. However, Gram-negative pathogens were more common in patients suffering from NMD compared with heart transplant patients (85.7% vs. 62.5%).

As evidenced by other studies from our institution, the most common MDRO/XDRO were carbapenem-resistant *Klebsiella pneumoniae* and *Acinetobacter baumannii* [34,35,36]. However, in this study, *Pseudomonas aeruginosa* was also a common pathogen.

Invasive devices were commonly used in our NMD patients. In fact, all the patients had at least one invasive device during hospitalisation. These data correlate with the literature describing invasive devices as risk factors for MDRO/XDRO infections [7,8].

In terms of treatment, it is interesting to note that isolates were mostly resistant to carbapenems but largely susceptible to colistin. As carbapenem resistance is not only a major actual concern but also an increasing future concern [37], the need for new antibiotics to cover for resistant Gram-negative pathogens remains important. In actual fact, new antibiotics have started showing in vitro resistance [38,39] and old effective antibiotics, such as colistin, have major side effects such as nephrotoxicity [40]. Furthermore, in our cohort of patients, even though colistin maintained a good antimicrobial activity against most isolates, it was not the most effective treatment option. Indeed, the three infectious syndromes treated with colistin did not translate into microbiological eradication. In fact, 38.9% of infectious syndromes were eradicated, a lower rate compared with a previous study on heart transplant recipients with MDRO infections, where an eradication rate of >50% was shown. Regarding new antibiotics, one of two cases treated with Cefiderocol eradicated the infection. In reality, the efficacy of Cefiderocol was found to be 66% in treating infections due to carbapenem-resistant Gram-negative bacteria [41].

None of the patients with MDRO/XDRO infection died during the hospitalisation. In contrast, one patient with colonisation died, but death was unrelated to infection. However, studies suggest a mortality rate among MDRO/XDRO of up to 36% [2,42]. Colonisation was previously found to be a risk factor for infection development [43]. In our study, among the 10 patients infected by an MDRO/XDRO strain, only one was previously colonised by the same pathogens.

Our study has limitations. First, it was a retrospective study; therefore, some data were missing, such as the eradication of infection in some patients. Due to the relatively low number of patients included, we could not assess the drivers of resistance. As this study was conducted at a single institution, the results may not be applicable to other settings with different local epidemiology. In addition, we did not assess data regarding the ratio of infection in this subgroup of patients compared with other patients with similar comorbidities such as neurologic sequelae after stroke, chronic respiratory diseases, or other patients hospitalized in intensive care units or other long-term inpatients.

In conclusion, MDRO/XDRO infection in patients affected by NMD are common, with Gram negatives prevailing. These polymorbid patients are vulnerable to this type of infection due to the characteristics of the disease requiring, in most cases, invasive devices, frequent hospitalisation, and antibiotic therapy. Since treatment options available for MDRO/XDRO infections are always affected by resistance development, we highlight the importance of infection prevention in this vulnerable population.

## Figures and Tables

**Figure 1 antibiotics-11-01411-f001:**
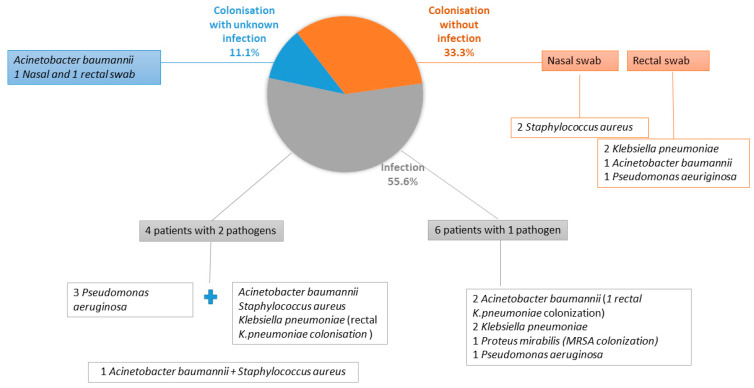
Distribution of colonisation without infection/MDRO or XDRO infection/colonization with unknown infection along with pathogens responsible, in the study cohort.

**Table 1 antibiotics-11-01411-t001:** Baseline characteristics of patients included in the study.

Parameter	All Patients Studied N. 18	MDRO/XDRO Infection	Colonisation without Infection
		N. 10	N. 6
Age, years	54 [45–69]	62 [43–70]	51 [43–61]
Sex M/F	10 (55.6)/8 (44.4)	3 (30)/7 (70)	5 (83.3)/1 (16.7)
Previous hospitalisation	12 (75)	7 (70)	3 (50)
Neuromuscular disorder			
Amyotrophic lateral sclerosis	14 (77.8)	8 (80)	4 (66.7)
Duchenne muscular dystrophy	1 (5.55)	1 (10)	-
Steinert myotonic dystrophy	1 (5.55)	1 (10)	-
Spinocerebellar ataxia type 2	1 (5.55)	-	1 (16.7)
Dysmetabolic axonal sensory-motor polyneuropathy	1 (5.55)	-	1 (16.7)
Comorbidities			
Congestive heart failure	2 (11.1)	1 (10)	0
Chronic kidney disease	2 (11.1)	1 (10)	0
Diabetes mellitus	2 (11.1)	1 (10)	1 (16.7)
Chronic pulmonary disease	1 (5.55)	0	1 (16.7)
Chronic liver disease	1 (5.55)	1 (10)	0
Peripheral vascular disease	1 (5.55)	1 (10)	0
Malignant neoplasia	2 (11.1)	2 (20)	0
Cerebrovascular accident	2 (11.1)	1 (10)	1 (16.7)
Hemiplegia	15 (83.3)	8 (80)	5 (83.3)
Charlson comorbidity index	3.5 [2–5.2]	4.5 [2–6]	3 [2–4.7]
Invasive procedures/devices	18 (100)	10 (100)	6 (100)
Parenteral nutrition	14 (77.8)	8 (80)	4 (66.7)
Peripherally inserted central catheter	9 (50)	5 (50)	2 (33.3)
Midline	1 (5.5)	1 (10)	0
Central venous catheter	3 (16.7)	2 (20)	0
Peripheral venous catheter	9 (50)	6 (60)	2 (33.3)
Tracheostomy	13 (72.2)	7 (70)	4 (66.7)
*Hematochemical data*			
Creatinine baseline, mg/dl	0.2 [0.15–0.45]	0.2 [0.15–0.45]	0.4 [0.4–0.4]
Creatinine onset 1 *, mg/dl	0.2 [0.2–0.5]	0.2 [0.1–0.45]	0.3 [0.2–1]
Creatinine onset 2 **, mg/dl	0.2 [0.1–0.3]	0.15 [0.1–0.45]	
Bilirubin baseline, mg/dl	0.6 [0.5–1.8]	0.67 [0.48–2.41]	0.62 [0.62–0.62]
Bilirubin onset 1 *, mg/dl	0.5 [0.4–0.7]	0.8 [0.3–2.1]	0.57 [0.46–0.68]
Bilirubin onset 2 **, mg/dl	0.6 [0.4– -]	0.49 [0.49–0.49]	
WBC > 15.000/mmc during hospital stay	6 (33.3)	4 [40]	0
PLT < 10.000/mmc during hospital stay	0	0	0
Albumin < 3.5 g/dl during hospital stay	6 (33.3)	5 (50)	0
Outcome			
Survived	17 (94.4)	10 (100)	5 (83.3)
Deceased	1 (5.6)	0	1 (16.7)

Data are N (%) or median [IQR]. * onset of first pathogen, ** onset of second pathogen.

**Table 2 antibiotics-11-01411-t002:** Characteristics of MDRO/XDRO-infected patients.

Parameter	Number (%) or Median [IQR]
Patients with infectious syndromes	10
Colonisation	3 (30)
Rectal swab positive	2 (20) (*K. pneumoniae*)
Nasal swab positive	1 (10) (MRSA)
Patients with colonisation who developed infection with the same pathogen	1 (10)
Patients with >1 pathogen	4 (40)
Total infectious syndromes	18
Types of infectious syndromes	
Pneumonia	9 (50)
Hospital-acquired pneumonia	7 (77.8)
Community-acquired pneumonia	1 (11.1)
Unknown	1 (11.1)
Complicated urinary tract infection	4 (22.2)
Bloodstream infection	2 (11.1)
Skin and soft tissue infection	3 (16.7)
Days from hospital entry to first isolation	24 [1–40]
Pathogens *Acinetobacter baumannii* *Pseudomonas aeruginosa* *Klebsiella pneumoniae* *Proteus mirabilis* *Staphylococcus aureus*	Total 14 pathogens 4 (28.6) 4 (28.6) 3 (21.4) 1 (7.1) 2 (14.3)
Eradication of infectious syndromes Yes/No/Missing data	7 (38.9)/8 (44.4)/3 (16.7)
Antibiotic	
Cefiderocol	2 (20)
Gentamicin	2 (20)
Ceftriaxone	3 (30)
Meropenem	1 (10)
Teicoplanin	2 (20)
Colisitin i.v	1 (10)
Colistin nebulised	2 (20)
Tigecycline	2 (20)
Ampicillin/Sulbactam	1 (10)
Piperacillin/Tazobactam	1 (10)
Linezolid	1 (10)
Ceftobiprole	1 (10)
Duration of therapy	12 [7–21]

## Data Availability

The dataset used for this study is available upon request to the corresponding author.

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
