# Peer review of "Clinical Characteristics and Outcome of MDR/XDR Bacterial Infections in a Neuromuscular Semi-Intensive/Sub-Intensive Care Unit"

_antibiotics, 2022, doi:10.3390/antibiotics11101411_

Round 1

Reviewer 1 Report

Thank you for the opportunity to review the manuscript “Clinical characteristics and outcome of MDR/XDR bacterial infections in a neuromuscular semi-intensive/Sub-intensive care unit (Manuscript number antibiotics-1938815)”.  

In this work, authors have tried to investigate the clinical and microbiological characteristics of MDRO infections in 18 patients who were admitted to a neuromuscular semi-intensive/sub-intensive care unit. They found that MDRO/XDRO infections in patients affected by NMD are common, with Gram negatives prevailing. The finding in itself is not new, and it is difficult to say that there is a unique characteristic in patients with NMD that distinguished it from other chronic ill patients. In a previous study of 47 heart transplant recipients admitted to another intensive care unit in the same hospital, the infection rate by MDRO/XDRO was 29.8%, and Gram-negative bacteria were the most prevalent etiological agents accounting for 62.5% of MDRO/XDRO isolates (Karruli A et al., Microorganism. 2021;9(6):1210). These findings appear similar despite the different host factors.

The following issues should be explained for the publication.

1) Could you prove that MDRO/XDRO infections occur more frequently in patients with NMD compared to other critically ill patients or patients with decreased respiratory drive (such as patients with severe neurologic sequelae after stroke, or patients with chronic respiratory diseases)?

2) How many patients with NMD were admitted to the neuromuscular semi-intensive/sub-intensive care unit during the study period, and what percentage of the patient with MDR/XDR isolated? Can it said that this ratio is high compared to other intensive care units or other long-term inpatients?

3) This study was performed in a single-center study. How can we distinguish if these findings are the characteristics of the patient with NMD or the epidemiologic characteristics of this center?

Last, please double-check the reference numbered list. After [20], they do not match the preceding sentence. 

Author Response

Reply to Reviewer 1:

Thank you for the opportunity to review the manuscript “Clinical characteristics and outcome of MDR/XDR bacterial infections in a neuromuscular semi-intensive/Sub-intensive care unit (Manuscript number antibiotics-1938815)”.  

In this work, authors have tried to investigate the clinical and microbiological characteristics of MDRO infections in 18 patients who were admitted to a neuromuscular semi-intensive/sub-intensive care unit. They found that MDRO/XDRO infections in patients affected by NMD are common, with Gram negatives prevailing. The finding in itself is not new, and it is difficult to say that there is a unique characteristic in patients with NMD that distinguished it from other chronic ill patients. In a previous study of 47 heart transplant recipients admitted to another intensive care unit in the same hospital, the infection rate by MDRO/XDRO was 29.8%, and Gram-negative bacteria were the most prevalent etiological agents accounting for 62.5% of MDRO/XDRO isolates (Karruli A et al., Microorganism. 2021;9(6):1210). These findings appear similar despite the different host factors. 

R: We thank the Reviewer for raising this important issue. We agree that NMD patients are at risk for developing MDRO/XDRO infection similar to other patients suffering from chronic illnesses. Transplant recipients are a different subtype of patients. Our aim was to provide new knowledge on a poorly studied clinical setting, i.e. that of patients with NMD in an advanced stage requiring sub-intensive care management. We are not claiming these patients have ‘unique’ characteristics. Regarding the type of pathogens, even though Gram-negatives prevailed in NMD patients as did also in heart transplanted patients, their prevalence was higher in the former compared to the latter (85.7% vs 62.5%). We have now added this information in the Discussion.

Page 7, lines 231-235: “In our experience, MDRO/XDRO infections were a common complication also in other subgroups of patients suffering from chronic illness such as heart transplanted patients with Gram-negative pathogens prevailing as published by our center [35]. However, Gram-negative pathogens were more common in patients suffering from NMD compared to heart transplanted patients (85.7% vs 62.5%).”

The following issues should be explained for the publication. 

1) Could you prove that MDRO/XDRO infections occur more frequently in patients with NMD compared to other critically ill patients or patients with decreased respiratory drive (such as patients with severe neurologic sequelae after stroke, or patients with chronic respiratory diseases)? 2) How many patients with NMD were admitted to the neuromuscular semi-intensive/sub-intensive care unit during the study period, and what percentage of the patient with MDR/XDR isolated? Can it said that this ratio is high compared to other intensive care units or other long-term inpatients? 

R: We thank the Reviewer for these comments. We agree that a comparison between this group of patients and other patients stated by the Reviewer and also the percentage of patients hospitalized with NMD who were affected by MDRO/XDRO infection could have been of great interest to the readers however, we did not collect such data since our aim was to describe the clinical characteristics in NMD patients with MDRO/XDRO. However, we have added this to the limitations of our study.

Page 8, lines 267-270: “Also, we did not assess data regarding the ratio of infection in this subgroup of patients compared to other patients with similar comorbidities such as neurologic sequelae after stroke, chronic respiratory diseases, or other patients hospitalized in intensive care units or other long-term inpatients.”

3) This study was performed in a single-center study. How can we distinguish if these findings are the characteristics of the patient with NMD or the epidemiologic characteristics of this center?

R: We thank the Reviewer for this comment. We agree that the epidemiological characteristics of the hospital could have influenced this study’s findings in particular prevailing pathogens. We mentioned this issue in the limitations of the study however to give a more detailed answer to the Reviewer’s suggestion we compared the prevailing pathogens in this article with other articles published by our institution.

Page 7, lines 236-238: “As evidenced by other studies from our Institution, the most common MDRO/XDRO were carbapenem-resistant Klebsiella pneumoniae and Acinetobacter baumannii [34-36]. However, in this study, also Pseudomonas aeruginosa was a common pathogen.”

Last, please double-check the reference numbered list. After [20], they do not match the preceding sentence. 

R: We thank the Reviewer for this suggestion. We double-checked and corrected the reference list.

Reviewer 2 Report

Remarks: 1. the discussion of the results is mainly based on the general variables (sex, age, etc.); 2. the component of microbiological diagnosis, antibiotic therapy, epidemiology, which represents the fundamental theme of the article, is not accompanied by appropriate scientific discussions, being reduced to a simple enumeration; 3. the numerical percentage statistical analysis, in a small number of cases, is irrelevant and can distort the results of the study; 4. the content of the tables has some inconsistencies that require reverification of the data and correlation of tables 2 and S1, so as to eliminate the elements of confusion:                           a) table 2 - patients with infectious syndrome – 10;                                    and in the following paragraph: total infectious syndromes – 18;                            b) table 2 – 14 identified pathogenic germs were isolated;                                    S1 – 12 identified germs subjected to phenotypic analysis; - c) 19 antimicrobials were used to treat 10 infections with 14 identified germs, details about the schemes used (monotherapy/combinations) are missing                            d) colistin did not prove to be the most effective, but retains its good antibacterial activity, because it was only used in 3 patients;                            e) the following figures must be commented and possibly correlated:                                 cure rate – 38.9%                                 persistence of infection - 45%                                1 death unrelated to the infection.   Recommendations: 1. renewing the discussion on the results of the study, deepening the information contained in the 3 tables; 2. the discussions must be focused mainly on microbiological diagnosis, infection, colonization, measures required to prevent infectious diseases of this category of patients and especially on the use of antimicrobial therapy;

Author Response

Reply to Reviewer 2:

Remarks: 1. the discussion of the results is mainly based on the general variables (sex, age, etc.); 2. the component of microbiological diagnosis, antibiotic therapy, epidemiology, which represents the fundamental theme of the article, is not accompanied by appropriate scientific discussions, being reduced to a simple enumeration;

Recommendations: 1. renewing the discussion on the results of the study, deepening the information contained in the 3 tables; 2. the discussions must be focused mainly on microbiological diagnosis, infection, colonization, measures required to prevent infectious diseases of this category of patients and especially on the use of antimicrobial therapy;

R: We thank the Reviewer for the suggestions. We expanded the Discussion of the article regarding the remarks and recommendations of the Reviewer. In particular, we more extensively addressed the issues of epidemiology, infection, colonisation, measures of prevention and treatment, as per Reviewer 1 request too. We also added new references. Specifically:

Page 7, lines 231-238: “In our experience, MDRO/XDRO infections were a common complication also in other subgroups of patients suffering from chronic illness such as heart transplanted patients with Gram-negative pathogens prevailing as published by our center [35]. However, Gram-negative pathogens were more common in patients suffering from NMD compared to heart transplanted patients (85.7% vs 62.5%)

As evidenced by other studies from our Institution, the most common MDRO/XDRO were carbapenem-resistant Klebsiella pneumoniae and Acinetobacter baumannii [34-36]. However, in this study, also Pseudomonas aeruginosa was a common pathogen.”

Page 8, lines 248-262: “Also, in our cohort of patients, even though colistin maintained a good antimicrobial activity against most isolates, it was not the most effective treatment option: indeed, the three infectious syndromes treated with colistin did not translate into microbiological eradication. In fact, 38.9% of infectious syndromes were eradicated, a lower rate compared to a previous study on heart transplant recipients with MDRO infections, where an eradication rate of >50% was shown. Regarding new antibiotics, one of two cases treated with Cefiderocol eradicated the infection. In fact, efficacy of Cefiderocol was found to be 66% in treating infections due to carbapenem-resistant Gram negative bacteria [41].

None of the patients with MDRO/XDRO infection died during the hospitalization. In contrast, one patient with colonization died, but death was unrelated to infection. However, studies suggest a mortality rate among MDRO/XDRO up to 36% [2,42]. Colonization was previously found to be a risk factor for infection development [43]. In our study, among the 10 patients infected by an MDRO/XDRO strain, only one was previously colonized by the same pathogens.

Methods, page 3 lines 111-113: “For infection prevention, in our center, we use chlorhexidine bathing along with substitution and sterile handling of catheters.”

  1. the numerical percentage statistical analysis, in a small number of cases, is irrelevant and can distort the results of the study;

R: We thank the Reviewer for pointing out this issue.  We have now changed from percentages to absolute numbers in the Results, in order to avoid any risk of data distortion.

  1. the content of the tables has some inconsistencies that require reverification of the data and correlation of tables 2 and S1, so as to eliminate the elements of confusion:a) table 2 - patients with infectious syndrome – 10;  and in the following paragraph: total infectious syndromes – 18;                    

R: We thank the Reviewer for highlighting this concern. However, in this case, the number is not mistaken as 10 patients had indeed 18 infectious syndromes, meaning that in one patient more than one type of infection in the same or different timeframe happened and that is why the numbers do not match exactly. We have specified this in the study Methods (page 3, lines 104-107).       

  1. b) table 2 – 14 identified pathogenic germs were isolated;S1 – 12 identified germs subjected to phenotypic analysis; 

R: We thank the Reviewer for highlighting this concern as well. However, in Table S1, the 12 identified germs subjected to the phenotypic analysis mentioned by the Reviewer are only the gram-negative isolates described on page one of the excel file of Table S1. On the second page of this file please find the 2 gram-positive isolates.

-c) 19 antimicrobials were used to treat 10 infections with 14 identified germs, details about the schemes used (monotherapy/combinations) are missing    

R: We thank the Reviewer for the suggestion. We have now added this information in the Results part. Page 7, lines 191-193: “Combination therapy was used in 3 patients, specifically: Ampicillin/Sulbactam + Gentamicin; Cefiderocol + Teicoplanin; Piperacillin/Tazobactam + Ceftriaxone. Other antibiotics were used in monotherapy.”

  1. d) colistin did not prove to be the most effective, but retains its good antibacterial activity, because it was only used in 3 patients; 

R: We thank the Reviewer for the suggestion. In fact, the three infectious syndromes which were treated with colistin did not eradicate infection. We added this information in the Results part page 6 lines 188-190: “Most of the isolates were susceptible to colistin. However, the three infectious syndromes treated with colistin did not eradicate the infection.” and also in the Discussion page 8, lines 248-251 “Also, in our cohort of patients, even though colistin maintained a good antimicrobial activity against most isolates, it was not the most effective treatment option: indeed, the three infectious syndromes treated with colistin did not translate into microbiological eradication.”                

  1. e) the following figures must be commented and possibly correlated:  cure rate – 38.9%, persistence of infection - 45%  1 death unrelated to the infection.   

R: We thank the Reviewer for the suggestion. We further discussed this issue in the Discussion part.

Page 8, lines 251-254: “In fact, 38.9% of infectious syndromes were eradicated, a lower rate compared to a previous study on heart transplant recipients with MDRO infections, where an eradication rate of >50% was shown.”

Page 8, lines 257-259: “None of the patients with MDRO/XDRO infection died during the hospitalization. In contrast, one patient with colonization died, but death was unrelated to infection. However, studies suggest a mortality rate among MDRO/XDRO up to 36% [2,42].

Round 2

Reviewer 1 Report

Your paper reads better now. 

Reviewer 2 Report

Now it is better.